# Establishing a Xanthan Gum–Locust Bean Gum Mucus Mimic for Cystic Fibrosis Models: Yield Stress and Viscoelasticity Analysis

**DOI:** 10.3390/biomimetics10040247

**Published:** 2025-04-17

**Authors:** Rameen Taherzadeh, Nathan Wood, Zhijian Pei, Hongmin Qin

**Affiliations:** 1Department of Biology, Texas A&M University, College Station, TX 77840, USA; woodn@tamu.edu; 2Department of Industrial & Systems Engineering, Texas A&M University, 3127 TAMU, College Station, TX 77843, USA; zjpei@tamu.edu

**Keywords:** mucus rheology, viscoelasticity, cystic fibrosis, yield stress, biomimetic mucus

## Abstract

Airway mucus plays a critical role in respiratory health, with diseases such as cystic fibrosis (CF) being characterized by mucus that exhibits increased viscosity and altered viscoelasticity. In vitro models that emulate these properties are essential for understanding the impact of CF mucus on airway function and for the development of therapeutic strategies. This study characterizes a mucus mimic composed of xanthan gum and locust bean gum, which is designed to exhibit the rheological properties of CF mucus. Mucus concentrations ranging from 0.07% to 0.3% *w*/*v* were tested to simulate different states of bacterial infection in CF. Key rheological parameters, including yield stress, storage modulus, loss modulus, and viscosity, were measured using an HR2 rheometer with strain sweep, oscillation frequency, and flow ramp tests. The results show that increasing the concentration enhanced the mimic’s elasticity and yield stress, with values aligning with those reported for CF mucus in pathological states. These findings provide a quantitative framework for tuning the rheological properties of mucus in vitro, allowing for the simulation of CF mucus across a range of concentrations. This mucus mimic is cost-effective, readily cross-linked, and provides a foundation for future studies examining the mechanobiological effects of mucus yield stress on epithelial cell layers, particularly in the context of bacterial infections and airway disease modeling.

## 1. Introduction

Mucus in the lungs serves as a critical protective barrier, trapping inhaled pathogens and particulates while facilitating their clearance through mucociliary transport [1]. This process, driven by coordinated ciliary motion and proper mucus hydration, is essential for maintaining airway health and preventing infection [2]. The epithelial cell layer of the lungs is continuously subjected to mechanical forces through coughing, inhalation, exhalation, and the movement of mucus across its surface. In pathologies such as cystic fibrosis (CF), the mucus layer that rests upon the epithelial surface becomes significantly thicker due to the disruption of ion exchange at the cell surface, particularly with chloride ion channel silencing. This leads to the production of highly viscous mucus that is difficult to clear, resulting in frequent bacterial infections, a hallmark of cystic fibrosis symptoms [3].

The field of mechanobiology has become increasingly important in understanding how mechanical forces influence cellular behavior and tissue function, particularly in the context of mucus-related diseases like cystic fibrosis (CF). Mucus, a complex biological hydrogel, plays a critical role in maintaining the integrity of epithelial barriers across various tissues, including the airways and gastrointestinal tract. Recent studies have highlighted the role of mechanical forces, such as shear stress and membrane distension, in modulating the behavior of cells exposed to mucus. For example, it has been shown that airflow-induced shear stress significantly influences mucin secretion in airway epithelial cells, with changes in the cytoskeleton playing a pivotal role in this process [4]. Similarly, studies on the gastrointestinal tract reveal that mechanical stresses, including shear stress and fluid flow, can regulate mucus production and barrier function in cell-based models, with shear-induced mucus production improving the physiological relevance of these models [5]. In CF, the mechanical properties of mucus and the airway environment are particularly critical, as the mutation of the CFTR gene leads to altered mucus viscosity and elasticity, contributing to impaired mucociliary clearance and chronic lung disease. Research has shown that CFTR is mechanosensitive, responding to mechanical stimuli such as shear stress and stretch, which may further influence the pathophysiology of CF [4].

One way to characterize CF pathology is by analyzing the rheological properties of airway mucus, particularly its yield stress, which is the minimum stress required to initiate flow. Yield stress provides critical insight into mucus clearance efficiency, as higher values indicate increased mucus tenacity and impaired transportability [6,7,8]. Elevated yield stress in diseased mucus compared with healthy mucus suggests that it could serve as a diagnostic marker for certain respiratory conditions. Additionally, yield stress measurements are increasingly being used to assess disease severity and to help classify bacterial infections present in the airway, with common bacterial infections and their associated yield stresses shown in Table 1 [9,10]. In the case of CF, bacterial biofilms, particularly those formed by Pseudomonas aeruginosa, contribute to elevated yield stress. The extracellular polymeric substances (EPSs) produced by these biofilms, largely composed of polysaccharides such as Psl and alginate, increase the mechanical toughness and intercellular cohesion of the biofilms, making them harder for immune cells to clear [6,11]. This highlights the relevance of studying mucus yield stress in the context of bacterial biofilm formation, as both factors contribute to impaired mucus clearance in CF patients. By tuning the viscoelastic properties of mucus mimics, we can more accurately model these biofilm-related mechanical properties and better understand their impact on epithelial cell behavior and mucus transportability in CF.

The development of 3D bioprinting and other in vitro modeling systems has also prompted the need for accurate mimics of the native lung environment, including both the epithelial layer and the mucus layer resting upon it. Many existing mucus mimics, such as pig gastric mucin (PGM)-based formulations, require cross-linkers like glutaraldehyde to achieve stable rheological properties [4,13,14,15,16]. These cross-linking methods introduce artificial covalent modifications that are highly variable and difficult to control, making them unreliable for studying mechanobiological effects on the epithelial layer. Similarly, alginate-based hydrogels rely on calcium ion cross-linking, which, while technically reversible with chelating agents such as EDTA, remains challenging to modulate dynamically in a biological setting. This limits precise control over viscosity and elasticity, particularly in applications requiring tunable mechanical properties without disrupting the system with external chelators [17,18]. These limitations hinder the reproducibility and adaptability of reported mucus models. Xanthan gum (XG) and locust bean gum (LBG) are polysaccharides that are widely studied for their rheological properties and synergistic behavior in gel formation. Xanthan gum, an anionic heteropolysaccharide (4171 KDa), forms weakly cross-linked networks in solution, while locust bean gum, a galactomannan (1054 KDa), enhances these networks through synergistic interactions [19,20]. The xanthan gum–locust bean gum (XG-LBG) system enables fine-tuning of viscoelastic properties solely through polymer concentration adjustments, eliminating the need for additional cross-linkers and increasing measurement reliability. While the rheological properties of XG and LBG in hydrogel systems, including the 1:3 ratio used in this study, have been well documented, their application to human lung epithelial cells, particularly 16HBE14o- cells, has not been explored. Currently, there is no documented evidence of LBG and XG being applied directly to human lung epithelial layers in research or therapeutic contexts. While these polysaccharides have been explored in other biological applications, such as supporting the in vitro development of bovine oocytes, their interaction with human lung epithelial cells, particularly in the context of cystic fibrosis (CF), remains unexplored [21,22]. This study represents the first effort to assess the biocompatibility of XG-LBG hydrogels with 16HBE14o- cells and their potential to serve as models for both healthy and CF lung mucus.

In this study, yield stress measurements were systematically performed by altering the concentrations of XG and LBG at a 1:3 ratio to match previously established yield stress values of cystic fibrosis and healthy patient sputum. The 1:3 ratio of XG to LBG was chosen based on prior research; in particular, a 1997 study that extensively documented the synergistic rheological behavior of these polysaccharides at varying concentrations [20]. Identifying concentrations that align with these yield stress values allows for further applications of the formulation to study the effects of yield stress on the epithelial cell layer. The ability to fine-tune viscoelastic properties while maintaining a biomechanically relevant composition makes this system an ideal candidate for in vitro airway models.

The 16HBE14o- cell line, derived from human bronchial epithelial cells, is widely used in pulmonary research due to its ability to form tight junctions and exhibit characteristics similar to native airway epithelium [23,24]. This study incorporates 16HBE14o- cells to evaluate the biocompatibility of the mucus mimic and assess its effects on epithelial integrity.

This study aims to expand upon prior rheological investigations of XG-LBG by presenting the rheological parameters of the mimic which align with the behavior of mucus, and the concentration-dependent variations in yield stress behavior, a critical but previously undocumented parameter for these systems. Additionally, we confirm the biomechanical relevance of this mucus mimic by assessing its compatibility with 16HBE14o- cells, providing a foundation for future applications in lung disease modeling and tissue engineering.

## 2. Materials and Methods

### 2.1. Materials and Chemicals

The mucus mimic was prepared using a blend of xanthan gum (XG) and locust bean gum (LBG) in a 1:3 ratio to simulate the viscoelastic properties of airway mucus. XG (100 g, FCC-grade) and LBG (250 g, FCC-grade) were obtained from Spectrum Chemical MFG (New Brunswick, NJ, USA). Five different concentrations were prepared: 0.3%, 0.2%, 0.15%, 0.10%, and 0.07% (*w*/*v*) in deionized water. The 0.07% concentration represents the minimum viable concentration for rheometric analysis, as lower concentrations resulted in samples that were too fluid and difficult to measure accurately. This concentration allowed us to mimic the viscoelastic properties of healthy mucus or the least severe form of cystic fibrosis mucus. From this point, the concentration was progressively increased to achieve yield stress values that align with documented levels of cystic fibrosis bacterial infections.

For each preparation, the required amount of XG was weighed on an analytical balance (Mettler Toledo LA203E) in a weigh boat and slowly dispersed into a 50 mL beaker containing deionized water, which was continuously stirred on a hot plate set to 37 °C. The XG powder was added gradually to prevent clumping. Following complete dispersion, the required amount of LBG was weighed separately and tapped granule by granule into the stirring solution to minimize clumping. This step was crucial to prevent aggregation, which could affect the homogeneity of the gel and impact subsequent rheometric analysis. After complete dispersion, the mixture was heated to 70 °C for 5 min to enhance polymer interaction and optimize gel formation. Heating above the xanthan gum helix–coil transition (~55 °C) disrupts molecular aggregates, leading to a more uniform distribution and stronger gel network upon cooling. This process has been shown to improve viscosity synergy and gel strength in XG-LBG systems [25]. The solution was then allowed to cool to room temperature while stirring before being transferred to sterile containers for further use.

### 2.2. Rheometric Analysis

A gap of 1000 µm was maintained for all tests. Each sample was tested at a constant temperature of 25 °C after a 30 s soak time to ensure thermal equilibrium. Three rheological tests were performed to characterize the viscoelastic and flow properties of the mucus mimic: an oscillation frequency sweep, a flow ramp test, and a strain sweep test in line with standard characterization procedures described in several papers [4,10,26,27,28,29]. The 1000 μm gap was selected based on particle size analysis of the xanthan–locust bean gum mixture, which revealed a particle size distribution with a mean of 125 μm and a median of 94 μm.

#### 2.2.1. Oscillation Frequency Sweep

The oscillation frequency sweep was conducted to evaluate the viscoelastic properties of the mucus mimic over a range of frequencies. A constant strain of 1% was applied while varying the angular frequency from 200 to 0.1 rad/s. The storage modulus (G′) and loss modulus (G″) were recorded to assess the relative contributions of elasticity and viscosity.

#### 2.2.2. Flow Ramp Test

The flow ramp test examined the shear-dependent viscosity of the mucus mimic. Shear rate was increased from 0.001 s^−1^ to 100 s^−1^ under steady flow conditions over a 60 s duration to characterize the sample’s shear-thinning behavior. This test provided insight into how the mucus mimic responds to increasing deformation rates, simulating physiological conditions where mucus is subjected to varying flow forces.

#### 2.2.3. Strain Sweep Test

The strain sweep (oscillation amplitude) test determined the linear viscoelastic region (LVR) and the strain limits of the mucus mimic. The test was performed at a fixed angular frequency of 6 rad/s, with strain increasing from 0.1% to 500%. The storage modulus (G′) and loss modulus (G″) were recorded to evaluate how the material’s structural integrity changes with increasing deformation.

#### 2.2.4. Yield Stress Determination

Yield stress is a critical parameter in characterizing mucus rheology, as it defines the minimum stress required to initiate flow. Several approaches have been employed to estimate yield stress in viscoelastic fluids, including empirical power-law models [10,27,30,31,32] and stress ramp-based methods [30,33,34].

Yield stress can be estimated using rheological models such as the Herschel–Bulkley, Bingham plastic, and Casson models [10,27,30,31,32]. These models describe the relationship between shear stress and shear rate, providing an extrapolated yield stress value based on flow behavior. The Herschel–Bulkley model is commonly used for non-Newtonian fluids such as mucus. They require extensive data fitting and assumptions about the fluid’s behavior under shear, which may not fully capture the structural transition of mucus-like materials.

In this study, yield stress was determined experimentally using the stress ramp method, in which shear stress was gradually increased while monitoring viscosity [30,33,34]. Yield stress was identified as the point of maximum viscosity prior to a rapid decline, marking the transition from a solid-like to a flowing state. This method was selected because it provides a direct measurement of yield stress without requiring curve fitting or extrapolation, captures the real-time transition from an elastic to a fluid-like response, and aligns with previous rheological studies on biological hydrogels and CF sputum, ensuring comparability with published data [10,35].

The obtained empirical, reproducible yield stress values can be directly correlated with mucus transport properties, offering a framework for tuning mucus mechanics in vitro. All rheometric data collected using TRIOS TA software (Version 5.1, TA Instruments, New Castle, DE, USA) were exported and analyzed in GraphPad Prism (Version 10.4.1, GraphPad Software, San Diego, CA, USA). Figures were constructed using GraphPad Prism to visualize the data for publication.

### 2.3. Mesh Size

The mesh size (ξ) of a hydrogel or gel-like material, such as the mucus mimic, provides crucial insight into its internal network structure and its influence on diffusion processes. The mesh size is directly related to the degree of cross-linking within the polymer network and can be inferred from the plateau modulus (G_0_), which represents the gel’s resistance to deformation in the linear viscoelastic regime. Importantly, (G_0_) is frequency-independent and reflects the material’s behavior at long timescales, where the network’s structural properties dominate.

To determine the mesh size, we adopted the theoretical framework presented by Pacheco et al. to derive the relationship between (G_0_) and (ξ) based on the principles of Rubber Elasticity Theory [36]. Specifically, the Generalized Maxwell model was employed to characterize the material’s viscoelastic response, allowing us to determine the shear modulus after complete relaxation. This relaxed modulus corresponds to the shear modulus of an ideal rubber-like material, as described by Rubber Elasticity Theory, and can be related to the mesh size through Equation (1):(1)$$\xi ={\left(\frac{k_{B}T}{G_{0}}\right)}^{1/3}$$
where *k_B_* is the Boltzmann constant, *T* is the temperature (K), and *G*_0_ is the plateau modulus (Pascal). This method provides a quantitative estimate of the linear distance between cross-linking sites in the mucus mimic and can be related to how the gel network might affect the movement of molecules or particles within it. Three independent replicates were performed, and the mean value across these replicates was reported as the final mesh size, offering a robust estimate of the material’s structural properties.

### 2.4. Cell Culture

16HBE14o- cells were purchased from Sigma Aldrich (Product No. SCC150, St. Louis, MO, USA) in January 2024. Cells were cultured in complete Modified Eagle Medium (MEM) (Sigma Aldrich, M2279) supplemented with 2 mM L-glutamine, 10% (*v*/*v*) fetal bovine serum (FBS) (Corning, New York, NY, USA), penicillin (100 U/mL), and streptomycin (100 µg/mL). Cells were maintained at 37 °C with 5% CO_2_ and routinely passaged at approximately 80% confluence using 0.25% Trypsin-EDTA. All experiments were conducted using cells at passage 6 from the date of purchase. Cells were tested negative for mycoplasma contamination based on a lack of extranuclear fluorescence using Hoechst 33342 as a DNA stain [37].

### 2.5. Cell Viability

Cell viability was assessed using fluorescent DNA stains. Hoechst 33342 (Thermo Fisher Scientific, Waltham, MA, USA) should permeate all cells and bind to DNA, resulting in nuclei fluorescence. SYTOX Green (Thermo Fisher Scientific, Waltham, MA, USA) is impermeant to the intact membrane of living cells, and will only result in nuclei fluorescence in dead cells.

Cells monolayers at 80% confluence were exposed to the thin (0.1% *w*/*v*), thick (0.3% *w*/*v*), or no mucus layers and assessed for cell viability one, three, and five days post-exposure. Prior to each assessment, cells were exposed to Hoechst 33342 (1 μg mL^−1^) and SYTOX Green (1 μM) for 15 min in the dark. Imaging was accomplished using the Echo Revolution inverted microscope with a stage top incubator using a 20X/0.6 dry objective with the DAPI filter for total cell nuclei fluorescence, and the FITC filter for dead cell nuclei fluorescence. Five images were taken at random locations for all six replicates.

Cell viability was quantified with the StarDist (version 0.9.1) nuclei segmentation tool using the included 2D fluorescence training set [38,39,40]. Images representing the total number of cells from the DAPI filter and the number of dead cells from the FITC filter were separated, as demonstrated in Appendix A, Figure A1. Images iteratively underwent histogram equalization prior to processing using StarDist (with the software’s default thresholding parameters). Output cell counts for the total number of cells and number of dead cells were combined and cell viability was calculated as shown in Equation (2).(2)$$Viability(\%)=\left(1-\frac{Dead\;Cells}{Total\;Cells}\right)\times 100$$

Statistical analysis was performed in GraphPad Prism (version 10.4.1). First, data normality was confirmed using the Shapiro–Wilk test. Since not all conditions passed the normality test (*p* < 0.05), non-parametric tests were applied. The Kruskal–Wallis test was used to assess the effect of mucus condition (control, thin, thick) and time point (Days 0, 1, 3, 5) on cell viability. Statistical significance was determined at *p* < 0.05, with results represented as * (*p* < 0.05) and ** (*p* < 0.01).

## 3. Results

### 3.1. Particle Size Analysis

Particle size analysis of the xanthan–locust bean gum mixture revealed a mean particle size of 125 μm and a median of 94 μm. Rheological tests were performed at a gap of 1000 μm based on this particle size distribution. Larger gaps (e.g., 3000 μm) led to sample slippage and significantly increased variability in the rheological measurements, particularly in G’’ (loss modulus). Statistical analysis (F = 469.26, *p* = 0.0021) confirmed that the 1000 μm gap produced the most reliable data, as shown in Supplementary Table A1 and Figure A2 in Appendix A.

### 3.2. Oscillation Frequency Test

Oscillatory frequency sweep tests were conducted from 0.1 to 200 rad/s across five concentrations of the mucus mimic (0.07%, 0.10%, 0.15%, 0.2%, and 0.3%) to assess their viscoelastic behavior. The results are compiled in Figure 1.

At 0.07% concentration, the weakest gel-like properties were observed, with significantly lower moduli compared with higher concentrations. The maximum storage modulus (G′) reached 0.7747 Pa, indicating weak elastic properties. At higher frequencies, the loss modulus (G″) surpassed G′, reaching a maximum of 3.10 Pa, signifying a transition toward a more fluid-like response. This concentration exhibited the earliest crossover between G′ and G″, occurring at 31.69 rad/s, indicating that the material becomes predominantly viscous at lower frequencies and is more prone to deformation under oscillatory stress. At 0.10% concentration, G′ initially dominated at low frequencies, but a crossover occurred at 50.24 rad/s, where G″ exceeded G′, marking the transition to viscous behavior. The maximum G” recorded was 17.10 Pa, demonstrating increased resistance to flow compared with the 0.07% concentration. At 0.15% concentration, the crossover shifted further to 79.62 rad/s, showing that a higher angular frequency was required for the material to exhibit a liquid-like response. G′ at low frequencies reached a maximum of 5.92 Pa, and G″ peaked at 14.94 Pa after surpassing G′. While the maximum G″ was slightly lower than that of the 0.10% concentration, it remained markedly high, suggesting a balance between elasticity and viscous dissipation. At 0.2% concentration, G′ dominated throughout the entire frequency range tested, with no observed crossover. The initial G′ measured at low frequencies was 13.32 Pa, and the final G″ at the highest frequency tested was 7.10 Pa, lower than the previous concentrations. This indicates that as polymer concentration increases, the material becomes more rigid and exhibits stronger resistance to deformation, behaving predominantly as a gel rather than a viscoelastic fluid. At 0.3% concentration, G′ remained dominant over G″ across most frequencies. However, unlike the 0.2% concentration, a trend toward a crossover was observed at 126 rad/s, the highest crossover frequency recorded among all tested concentrations. The final G″ after surpassing G′ was 8.35 Pa, indicating that despite its strong gel-like properties, the material can transition to a more fluid-like state at sufficiently high frequencies. Across all concentrations, the crossover between G′ and G″ progressively required a higher angular frequency as the concentration increased, indicating that higher polymer content enhances elastic properties and delays the transition to a viscous-dominated state. At 0.2%, the mucus mimic exhibited a fully elastic response within the tested range, whereas at 0.3%, a delayed transition to viscous behavior was observed. These findings suggest that increasing polymer concentration shifts the mucus mimic toward a more gel-like, structurally stable material, with high concentrations displaying stronger resistance to deformation and prolonged elasticity.

### 3.3. Flow Ramp Test

The flow ramp test revealed that the mucus mimic exhibits shear-thinning behavior, a characteristic of non-Newtonian fluids where viscosity decreases with increasing shear rate. This property is crucial for mucus function, as it allows the material to maintain a protective gel-like consistency at low stress while becoming more fluid-like under higher shear conditions, such as during coughing or mucociliary transport. All concentrations were compiled into Figure 2 for graphical representation of linear sheer thinning with increasing concentration.

At the lowest shear rate (0.88 s^−1^), viscosity was highest for all concentrations, with 0.30% exhibiting the greatest resistance to flow (8.99 Pa·s), followed by 0.20% (5.35 Pa·s) and 0.15% (2.13 Pa·s). The lowest viscosity values were observed for 0.07% (0.59 Pa·s) and 0.10% (0.63 Pa·s), indicating that lower concentrations form weaker gels with less structural integrity. As shear rate increased, all concentrations demonstrated a decline in viscosity, confirming a shear-thinning response. This effect was most pronounced in the lower-concentration samples (0.07% and 0.10%), where viscosity rapidly decreased, reaching final values of 0.038 Pa·s and 0.075 Pa·s at 99 s^−1^, respectively. The higher-concentration formulations (0.20% and 0.30%) exhibited a more gradual decline, with viscosity stabilizing at 0.41 Pa·s and 0.86 Pa·s at the highest shear rate tested.

### 3.4. Oscillation Amplitude

The linear viscoelastic (LVE) region was identified for all mucus mimic concentrations as the strain range where the storage modulus (G′) remained relatively stable and higher than the loss modulus (G″), indicating that the material maintained its structure under small deformations. As strain increased, G′ and G″ began to move closer together, signaling the point at which the material started to respond more to the applied strain. The extent of this region varied across concentrations, with higher polymer concentrations maintaining stability over a wider strain range before G′ and G″ started to converge. All concentrations presented in Figure 3 show the convergent nature of the moduli at all concentrations.

At 0.07% concentration, the LVE region extended up to approximately 99.87% strain, where G′ was 0.703 Pa and G″ was 0.092 Pa. Beyond this point, the gap between the two began to decrease. The 0.10% concentration remained stable until 99.75% strain, where G′ was 1.907 Pa and G″ was 0.221 Pa, showing slightly stronger elasticity compared to the 0.07% sample. For the 0.15% sample, G′ remained higher than G″ up to 99.77% strain, where G′ reached 11.56 Pa while G″ was 1.29 Pa. In the 0.2% concentration, this behavior continued until 101.62% strain, where G′ was 9.88 Pa and G″ was 1.49 Pa. The 0.3% concentration had the longest LVE region, maintaining stable G′ values until 105.2% strain, where G′ was 19.77 Pa and G″ had increased to 5.11 Pa. Although G′ remained greater than G″ across all strain levels, the narrowing difference between them at higher strains suggests that the material responded more to applied deformation at these points. These results show that higher polymer concentrations extend the LVE region, meaning that these formulations can maintain their structure over a wider range of strain. The lower concentrations showed an earlier decrease in the gap between G′ and G″, meaning they reacted to strain sooner. The gradual approach of G′ and G″ with increasing strain provides insight into how polymer concentration affects the material’s ability to withstand deformation, which is relevant for understanding how mucus behaves under different physiological conditions.

### 3.5. Stress Ramp Test

The yield stress of the mucus mimic was determined from the time vs. viscosity test, where viscosity decreases over time as stress is applied. Yield stress represents the point at which the material transitions from a solid-like state to a flowing state. The results show a clear trend of increasing yield stress with higher polymer concentration, indicating that more concentrated formulations require greater stress to initiate flow. Figure 4 presents the viscosity response as a function of applied stress.

Table 2 with associated yield stresses from shows at 0.07% concentration, the yield stress was 1.095 Pa (±0.029 SEM), the lowest among all samples, indicating that this formulation required minimal stress to begin flowing. The 0.10% concentration exhibited a slightly higher yield stress at 2.184 Pa (±0.062 SEM), suggesting a small increase in structural resistance. A more substantial increase was observed at 0.15% concentration, where yield stress reached 5.714 Pa (±0.154 SEM), indicating a stronger gel network. The 0.20% and 0.30% formulations displayed the highest yield stress values, at 9.364 Pa (±0.232 SEM) and 12.20 Pa (±0.293 SEM), respectively, confirming that these higher concentrations form more rigid structures that resist flow significantly before yielding. These findings align with the expected behavior of mucus-like materials, where lower concentrations behave more fluidly, while higher concentrations exhibit stronger gel-like properties [19].

### 3.6. Mesh Size

The mesh size of the mucus mimic was estimated by applying the method described by Pacheco et al., which uses the plateau modulus (G_0_) obtained from oscillatory frequency sweeps [36]. The results of the mesh size analysis are presented in Table 3, showing a clear trend of decreasing mesh size with increasing mucus mimic concentration. Specifically, at lower concentrations (0.07% *w*/*w*), the mesh size was estimated to be 249 ± 22 nm, while at higher concentrations (0.30% *w*/*w*), the mesh size reduced to 76 ± 2 nm.

### 3.7. Cell Viability

The viability assay confirmed that the mucus mimic had no significant effect on 16HBE14o- cell survival, indicating biocompatibility. Figure 5 presents quantified viability results across the minimum and the maximum range of XG-LBG concentrations examined in this paper. Data represent the mean ± SEM of six biological replicates (*n* = 6 wells per condition), with five images taken per well. The Kruskal–Wallis test was used to compare the viability between control, thin mucus, and thick mucus conditions at each time point (Day 0, Day 1, Day 3, Day 5). No significant differences in cell viability were observed between conditions (*p* > 0.05).

## 4. Discussion

This study aimed to develop and evaluate a rheologically controlled mucus mimic that emulates key mechanical properties of cystic fibrosis (CF) airway mucus. By focusing on three major rheological characteristics, shear thinning, linear viscoelastic behavior, and yield stress, this mucus mimic serves as a platform for studying how mucus mechanics, independent of biochemical factors, influence epithelial health and function. The mesh size of the mucus mimic was also estimated, with values ranging from 249 ± 22 nm at 0.07% concentration to 76 ± 2 nm at 0.30%, showing a decrease in mesh size with increasing concentration. These values are consistent with the mesh size reported for CF sputum, which ranges from 60 to 300 nm (average 140 ± 50 nm) [41], providing further validation of the model’s relevance to CF research. This mesh size alignment suggests that our mucus mimic not only emulates the rheological properties of CF mucus but also its structural features, enabling studies of diffusion and transport in a CF-like environment [42]

The results demonstrate that all tested concentrations of the mucus mimic exhibited shear-thinning behavior, similar to real mucus [28]. Higher-concentration formulations (0.2% and 0.3%) showed higher viscosity across all shear rates, suggesting that these concentrations could hinder mucus transport, mirroring the characteristics of CF mucus. The linear viscoelastic (LVE) region, where the mucus maintains its internal structure without significant breakdown, was also affected by concentration. At lower concentrations (0.07–0.10%), the LVE region was shorter, reflecting a more flexible material similar to the lower range of CF severity, which is more easily cleared. In contrast, at higher concentrations (0.2–0.3%), the LVE region was longer, indicating more rigid, gel-like behavior, comparable to *P. aeruginosa* presence [9]. This increased rigidity could exert mechanical stress on epithelial cells, potentially disrupting barrier integrity, reducing ciliary function, or triggering inflammatory responses [43]. The prolonged LVE region observed at higher concentrations suggests that the mucus mimic can effectively emulate the mechanical resistance of CF mucus and its impact on epithelial function. Yield stress, the minimum force required to transition the mucus from a solid-like state to a flowing liquid, was measured across the different concentrations. At 0.07% concentration, the yield stress was 1.095 Pa, aligning with the range observed for healthy mucus. At 0.3%, the yield stress increased to 12.20 Pa, which falls within the range of CF patient mucus with bacterial infection, as noted in Table 1. For example, in CF with Pseudomonas aeruginosa infection, yield stress can reach up to 20 Pa, indicating that the mucus mimic can emulate the mechanical properties of CF mucus during infection. These stress values closely mirror the progression of disease severity in CF, where mild disease exhibits around 2 Pa, while severe CF with bacterial infection can reach 20 Pa. The ability to tune the yield stress of this mucus mimic allows it to emulate the mechanical properties of mucus in bacterial infections such as those caused by *Pseudomonas aeruginosa* and *Staphylococcus aureus*, where increased viscosity and yield stress are observed [44]. This provides a relevant in vitro system to study how bacterial infections influence mucus biomechanics and their impact on the airway environment and epithelial cell behavior through air–liquid interface application. It has also been confirmed that the mucus mimic did not negatively impact the viability of 16HBE14o- cells across the tested concentration range (0.07% to 0.3%). This demonstrates that the mucus mimic is biocompatible and suitable for use in longer-term cell culture experiments, making it a valuable tool for mechanobiology studies.

While the mucus mimic successfully emulates key mechanical properties of native mucus, including shear-thinning behavior, viscoelasticity, and yield stress, it does not incorporate the full biochemical composition or biological functions intrinsic to native mucus. Therefore, while it is biomechanically relevant, its biological relevance is limited. The model lacks the mucins, lipids, antimicrobial peptides, and other bioactive molecules found in native mucus, which are crucial for its biological functions such as pathogen trapping, immune modulation, and mucosal protection [45]. Consequently, the absence of these bioactive components limits the model’s ability to emulate the full biological interactions seen in vivo, such as those between mucus, pathogens, and epithelial cells. As such, this model is better suited for studying the mechanical properties of mucus and their impact on epithelial cells, rather than simulating the full biological processes. Future studies should aim to integrate biochemical components to further enhance the physiological fidelity of this model. Additionally, future research could extend the mesh and pore size characterization of the mucus mimic using scanning electron microscopy (SEM) imaging. This would provide a more direct and detailed visualization of the network structure at the micro and nano-scales, offering valuable insight into how the mechanical and structural features of mucus affect diffusion and transport properties.

## 5. Conclusions

This mucus mimic demonstrates a range of rheological properties that effectively emulate the mechanical characteristics of CF mucus. By incorporating varying concentrations of XG and LBG, we have created a tool that can be used to explore the impact of impaired mucus clearance on epithelial cell behavior, barrier integrity, and inflammatory responses. Further studies are needed to investigate how these mechanical properties interact with the biochemical aspects of mucus in disease states. Future work should focus on studying the effects of the mucus mimic on ciliary function, tight junction integrity, and mechanobiological pathways in epithelial cells on the air–liquid interface, as well as further exploring how the mucus mimic can model the mechanical forces that drive disease progression in CF. 

## Figures and Tables

**Figure 1 biomimetics-10-00247-f001:**
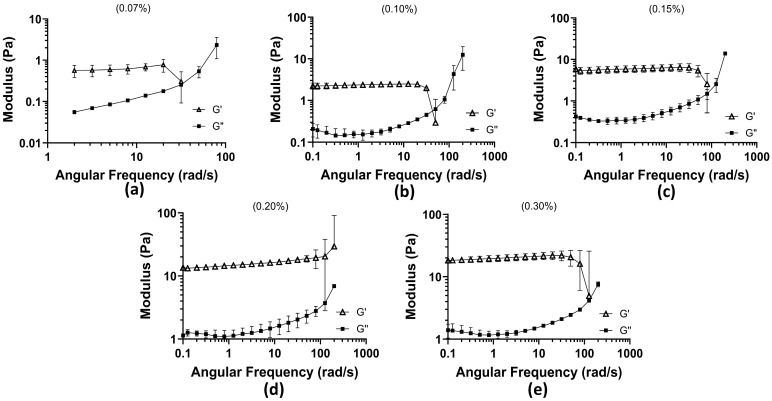
Oscillation frequency test results among 5 concentrations of 1:3 xanthan–locust bean gum: (**a**) 0.07% concentration with angular frequency (rad/s) plotted against Moduli (Pa) (the 0.1–1 rad/s range was cut out due to the rheometer operating at its limit; the data were not supportive of behavior); (**b**) 0.10% concentration; (**c**) 0.15% concentration; (**d**) 0.20% concentration; and (**e**) 0.30% concentration.

**Figure 2 biomimetics-10-00247-f002:**
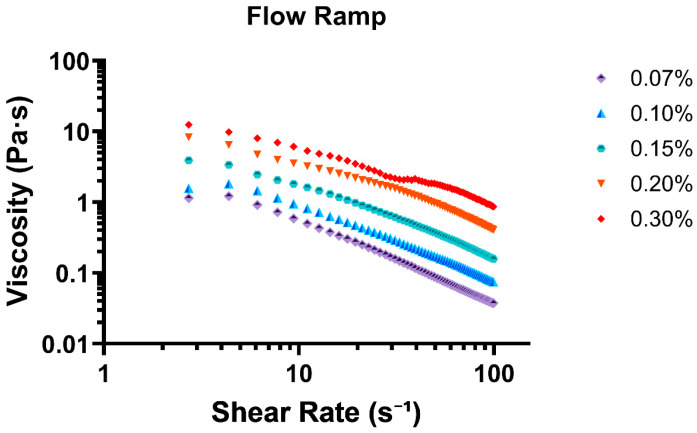
Flow ramp test displaying all 5 tested concentrations as shear rate (s^−1^) vs. viscosity (Pa·s).

**Figure 3 biomimetics-10-00247-f003:**
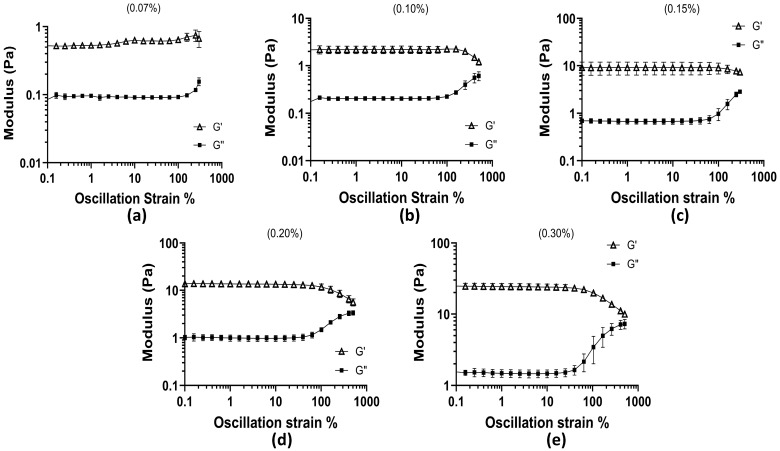
Oscillation amplitude test across all concentrations as oscillation strain % vs. moduli (Pa): (**a**) 0.07% concentration; (**b**) 0.10% concentration; (**c**) 0.15% concentration; (**d**) 0.20% concentration; and (**e**) 0.30% concentration.

**Figure 4 biomimetics-10-00247-f004:**
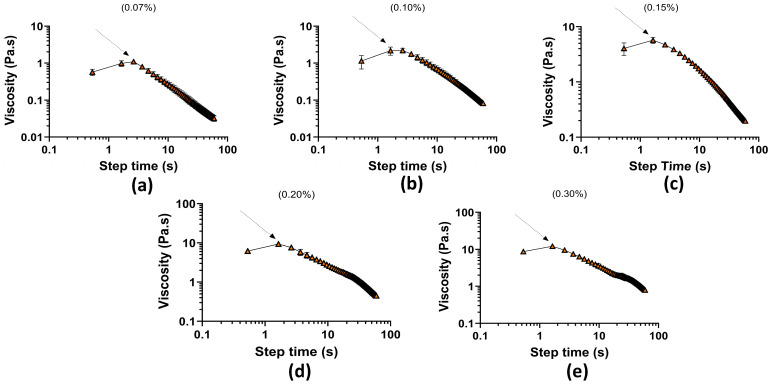
Stress ramp test as a function of time (s) vs. viscosity (Pa·s): (**a**) 0.07% concentration; (**b**) 0.10% concentration; (**c**) 0.15% concentration; (**d**) 0.20% concentration; and (**e**) 0.30% concentration.

**Figure 5 biomimetics-10-00247-f005:**
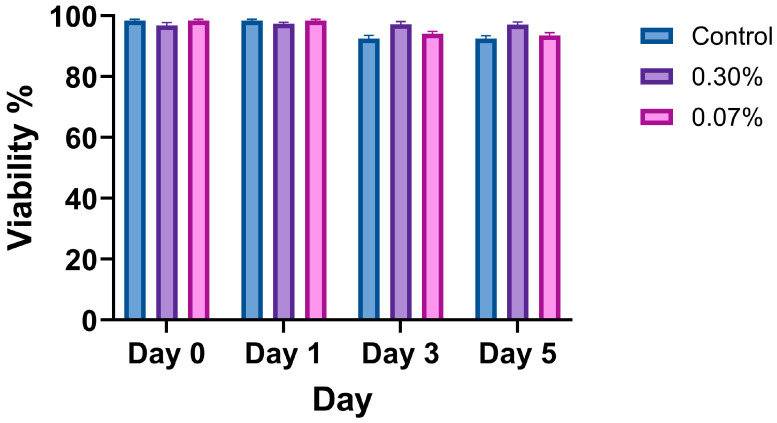
Viability of 16HBE14o- cells treated with control, thin mucus (0.07%), or thick mucus (0.3%) at days 0, 1, 3, and 5. Data reported as arithmetic mean ± standard deviation. Significance was determined using the Kruskal–Wallis test, and no significant differences between conditions were observed (*p* > 0.05).

**Table 1 biomimetics-10-00247-t001:** Common yield stress values for CF mucus.

Mucus Source	Condition	Yield Stress (Pa)	Reference
CF Severity	Mild	~2	[10,12]
	Severe	~9	
	Very Severe	~9	
CF with Bacterial Infection Type	*S. aureus*	~4	
	*S. maltophilia*	~10	[9,10,12]
	*B. cepacia*	~10	
	*P. aeruginosa*	~20	

**Table 2 biomimetics-10-00247-t002:** Yield stress results.

Concentration (%)	Yield Stress (Pa)	Standard Error (SEM)	Sample Size (*n*)
0.07%	1.095	0.02946	3
0.10%	2.184	0.06272	3
0.15%	5.714	0.1544	3
0.20%	9.364	0.2322	3
0.30%	12.20	0.2935	3

Yield stress values were determined from step time vs. viscosity plots (stress ramp). Yield stress measurements were performed in triplicate (*n* = 3).

**Table 3 biomimetics-10-00247-t003:** Mesh size results.

Concentration (% *w*/*w*)	Plateau Modulus, *G*_0_ (Pa)	Mesh Size, *ξ* (nm)
0.07	0.53 ± 0.16	249 ± 22
0.10	2.36 ± 0.28	118 ± 2
0.15	5.51 ± 0.80	114 ± 6
0.20	13.66 ± 0.48	84 ± 1
0.30	18.70 ± 1.53	76 ± 2

Data represent the mean ± standard deviation from three independent replicates. The mesh size (ξ) was calculated using the plateau modulus (G_0_) determined from the frequency-independent region of the storage modulus.

## Data Availability

Data supporting the findings of this study are available from the corresponding author upon reasonable request.

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
