# Peer review of "Establishing a Xanthan Gum–Locust Bean Gum Mucus Mimic for Cystic Fibrosis Models: Yield Stress and Viscoelasticity Analysis"

_biomimetics, 2025, doi:10.3390/biomimetics10040247_

Round 1

Reviewer 1 Report

Comments and Suggestions for Authors

General Comment

The authors describe the preparation and rheological analysis of a mucus-mimicking hydrogel composed of xanthan gum (XG) and locust bean gum (LBG) aimed at simulating cystic fibrosis mucus. The manuscript provides thorough rheological characterization; however, the biological significance and novelty of this model remain insufficiently demonstrated. Several claims made in the manuscript are not fully supported by the presented evidence and require revision. Detailed comments highlighting these issues are provided below.

Major Comments

  1. Mucus is a complex and heterogeneous biological mixture composed of mucin glycoproteins, lipids, salts, and diverse biochemical components. Particularly, mucin glycoproteins, with their complex and diverse O- and N-linked glycans, are central to mucus's biological function. In contrast, the carbohydrate structures present in xanthan gum and locust bean gum fundamentally differ from mucin glycans. Consequently, the proposed XG-LBG hydrogel does not mimic the biochemical composition and biological functions intrinsic to native mucus. The authors should delineate the limitations and adjust claims of biological relevance. At best, the presented model can be considered biomechanically relevant.
  2. The rheological properties of XG and LBG hydrogels, including the 1:3 ratio utilized in this manuscript, are already well documented in previous literature. The authors should explicitly state what is novel about their study and clearly differentiate their approach from existing reports.
  3. Related to the previous comment, several in vitro models mimicking both the biochemical and biomechanical properties of physiological and pathological mucus already exist (e.g., Pacheco et al., 2019, 10.1039/C9TB00957D; Boegh et al., 2014, 10.1016/j.ejpb.2014.01.001; Gyarmati et al., 2022, 10.1016/j.colsurfb.2022.112406). The manuscript lacks proper discussion regarding when or why the XG-LBG model would be preferable or advantageous over existing alternatives. The authors should provide a comparative table or section highlighting the distinct features, benefits, and potential limitations of their proposed model relative to existing mucus mimics.
  4. The manuscript lacks any structural characterization of the hydrogel matrix, which would help substantiate claims related to its microstructure and mechanical properties. Data on the mesh size or pore structure of the proposed mucus model (e.g., SEM, Cryo-EM) should be considered. Such data would significantly enhance the robustness and clarity of the presented results.
  5. A proper discussion of the model's limitations is not provided. Authors should explicitly discuss and clarify the limitations inherent in their model, including biochemical composition, absence of bioactive functions, and potential implications for its applicability in cystic fibrosis research or drug delivery.

Minor Comments

  1. Numerous referencing errors and inconsistencies are present throughout the manuscript. Authors should carefully revise and verify all references.
  2. Some results are improperly discussed in the Methods section (e.g., lines 130-135). All interpretations and discussions of experimental data should be placed in the Results and Discussion sections.
  3. Figure 1 should be moved to supplementary information.
  4. The authors mention that alginate-based hydrogels irreversibly gel upon calcium cross-linking (lines 63-65). This statement is misleading, as alginate crosslinking can indeed be reversed using chelators (e.g., EDTA). The authors should correct this point and clarify scenarios in which reversible gelation would offer distinct advantages.
  5. The manuscript lacks citations of relevant studies from the mucus and hydrogels fields. Authors should comprehensively review recent literature and ensure critical references are included to provide better context and strengthen the manuscript’s scientific rigor.

Reviewer 2 Report

Comments and Suggestions for Authors

The manuscript "Establishing a Xanthan Gum-Locust Bean Gum Mucus Mimis for Cystic Fybrosis Model" describes the rheological characterization and viability studies. The work is quite good, but some aspects that should be improved:

1. Introduction

  • Table 1 is not mentioned in the text and it is missing the units for Yield stress.

2. Methods

2.2 Rheometric analysis

  • Each rheological analysis should be separated into subitems 2.2.1, 2.2.2.
    -Besides, the results of the rheological analysis regarding the gaps between the geometries and the most suitable test to determine yield stress should be included in the Results and Discussion session.

2.4. Cell viability

  • Figure 1 should be placed in the Results session and the caption description should be improved with more details about the picture.

3. Results

  • The authors should include a Figure with the 5 mucus mimic systems before the rheological results.
  • Page 12. line 384. The authors discussed "The ability to tune the yield stress with this model provides a relevant in vitro system to study the biomechanical effects of bacterial infections on epithelial cells." Could you described how these studies could be performed? How does theses mucus mimic could help in theses evaluations?
Comments on the Quality of English Language

Some typos should be improved, regarding the name of the bacteria, for example P. aerugionsa, S. auereus, B. Cepacia. As these words are scientific names, they should be presented in italic.

Reviewer 3 Report

Comments and Suggestions for Authors

Manuscript entitled “Establishing a Xanthan Gum-Locust Bean Gum Mucus Mimic for Cystic Fibrosis Models: Yield Stress and Viscoelasticity Analysis” has been reviewed. Following points should be considered before further recommendation.

Abstract is quite simple and general. Modify the abstract in concise way. Provide background of study, major findings and conclusive lines in abstract.

Reduce the keywords

Why there is need of this study?

What is the novelty and scope of study?

Why authors have selected Xanthan Gum-Locust Bean Gum only? Why not other polysaccharides?

Section 2.1 should be materials and chemicals

Units should be carefully check in while manuscript. Use min instead of Minutes.

On the basis of what authors have selected this concentration for the preparation of mucus mimic?

Remove these lines 123-124

Results and discussion should be improve by providing suitable justifications with references.

What is the limitation of this study?

Comments on the Quality of English Language

Should be improved during revision. 

Round 2

Reviewer 1 Report

Comments and Suggestions for Authors

The authors has mostly answered the questions raised. The revised manuscript is improved after the inclusion of mesh-size analysis and discussion of the critical limitations of the model. I have no further comments.

Reviewer 3 Report

Comments and Suggestions for Authors

Authors have revised the manuscript accordingly.